# Effect of fixed 7.5 minutes' moderate intensity exercise bouts on body composition and blood pressure among sedentary adults with prehypertension in Western-Kenya

Karani Magutah[1]*, Grace Mbuthia[2], James Amisi Akiruga[3], Diresibachew Haile[1], Kihumbu Thairu[4]

1 Department of Medical Physiology, School of Medicine, Moi University, Eldoret, Kenya, 2 College of Health Sciences, Jomo Kenyatta University of Agriculture and Technology, Nairobi, Kenya, 3 Department of Family Medicine, Medical Education and Community Health, Moi University School of Medicine, Eldoret, Kenya, 4 Department of Medical Physiology, School of Medicine, University of Nairobi, Nairobi, Kenya

* kmagutah@mu.ac.ke, jkarani@cartarica.org, kmagutah@gmail.com

## Abstract

Prehypertension is a modifiable risk factor for cardiovascular disease observed to affect an estimated 25–59% of global population and closely associated with body composition. Without appropriate interventions, one-third of individuals with prehypertension would develop full-blown hypertension within 4 years. The existing exercise recommendations need substitutes that appeal more yet accord similar or better outcomes in desire to halt this progression. This study evaluated the effect of Fixed 7.5-minute Moderate Intensity Exercise (F-7.5m-MIE) bouts on Body Composition and Blood Pressure (BP) among sedentary adults with prehypertension in Western-Kenya in a Randomized Control Trial (RCT) performed throughout the day compared to the single-continuous 30-60-minute bouts performed 3 to 5 times weekly. This RCT, with three arms of Experimental Group1 (EG1) performing the F-7.5m-MIE bouts, Experimental Group 2 (EG2) performing current World Health Organization (WHO) recommendation of ≥30-min bouts, and, control group (CG), was conducted among 665 consenting pre-hypertensive sedentary adults enrolled from western Kenya. EG1 and EG2 performed similar weekly cumulative minutes of moderate intensity exercises. Adherence was determined using activity monitors and exercise logs. Data regarding demographic characteristics, heart rate, BP, and anthropometric measures were collected at baseline and 12th week follow-up. Data regarding univariate, bivariate and multivariate (repeated measurements between and within groups) analysis were conducted using STATA version 13 at 5% level of significance. The study revealed that males (92.1% in EG1, 92% in EG2 and 96.3% in CG) and females (94.6% in EG1, 89.3% in EG2 and 95% in CG) in the three arms completed the exercise at follow-up respectively. At 12th week follow-up from all exercise groups, males' and females' measurements for waist-hip-ratio, waist-height-ratio, systolic BP (SBP), heart rate and pulse pressure showed significant drops from baseline, while diastolic BP (DBP) and body mass index (BMI) reported mixed results for males and females from the various treatments. Both treatments demonstrated favourable outcomes. However, differences in the change between baseline and endpoint yielded

**Data Availability Statement:** Data have been availed with this submission.

**Funding:** KM. This research was supported by the Consortium for Advanced Research Training in Africa (CARTA). CARTA is jointly led by the African Population and Health Research Center and the University of the Witwatersrand and funded by the Carnegie Corporation of New York (Grant No-G-19-57145), Sida (Grant no. 54100113), Uppsala Monitoring Centre and the DELTAS Africa Initiative (Grant No: 107768/Z/15/Z). The DELTA Africa Initiative is an independent funding scheme of the African Academy of Sciences (AAS) Alliance for the Accelerating Excellence in Science in Africa (AESA) and supported by the New Partnership for Africa's Development Planning and Coordinating Agency (NEPAD Agency) with funding from Wellcome Trust (UK) and the UK government. The statement made and views expressed are solely the responsibility of the Fellow. The funders had no role in study design, data collection and analysis, decision to publish, or preparation of the manuscript.

**Competing interests:** The authors have declared that no competing interests exist.

mixed outcomes (SBP; $p < 0.05$ for both males and females, DBP; $p < 0.05$ for males and females, waist-height-ratio; $p = 0.01$ and $< 0.05$ for males and females respectively, waist-hip-ratio; $P = 0.01$ and $> 0.05$ for males and females respectively, BMI; $p > 0.05$ for both males and females, heart rate; $p < 0.05$ for males and females and pulse pressure; $p = 0.01$ and $> 0.05$ for males and females respectively). The study design however could not test for superiority. The study demonstrated that the F-7.5m- MIE treatment programme and the WHO recommended 3–5 times weekly bouts of 30–60 minutes regime produced comparably similar favourable outcomes in adherence and BP reductions with improved body composition.

**Trial registration**: Trial registered with Pan African Clinical Trial Registry (www.pactr. org): no. PACTR202107584701552. (S3 Text)

## Introduction

Sedentary lifestyles has contributed to the recent increase in non-communicable diseases in Kenya, causing a heavy health and economic burden [1]. Hypertension, the commonest cardiovascular disease (CVD) affects 1 in 7 people globally, contributing the most mortality [2–4]. Traditionally defined as blood pressure (BP) $\geq 140/90$, revised definition has included lower values $\geq 130/80$ [5]. A third (33%) of individuals with higher-normal BP develop hypertension with age [6, 7]. Identifying at-higher-risk individuals early and intervening before full-blown disease is critical [6–8]. Prehypertension, the pre-disease state, is defined as systolic BP (SBP) $\geq 120–139$ and/or diastolic BP (DBP) $\geq 80–89$ in individuals aged 18 years and above on $\geq 2$ consecutive measurements. Although recently the term "elevated" BP (SBP 120–129 and DBP $< 80$ mmHg) was proposed, these proposals are not yet adopted in Kenya [5–7, 9, 10] and literature based on this new definition lacks. Prehypertension affects 25–59% of the global population [6, 9]. Independently, it is a modifiable risk factor for CVD.

At 26% prevalence in sub-Saharan Africa (SSA), hypertension burden and relationship to CVD is a growing concern [11]. In 2018, 25% of Kenyans had hypertension, highlighting extent of the burden locally [4, 12]. The prevalence of prehypertension in SSA ranges from 21–33% [3, 11] and was 47% (51% for males; 46% for females) in Kenya in 2018 [4], higher than neighbouring Uganda (33%; 42% for males, 29% for females) [13]. Prevalence peaks at 69 years, dropping thereafter coinciding with increasing hypertension prevalence as individuals transition to full-blown hypertension [4, 7]. Without interventions, 20–33% of individuals with prehypertension develop hypertension within 4 years especially if they have higher DBP before age 50, and the risk doubles at BP ranges 130-139/85-89 mmHg as opposed to 120-129/ 80-84 mmHg [6, 14, 15].

Pre-hypertension also poses direct risk to CVD by associating with chronic cardiac and vascular changes like arterial stiffness and decreasing intima-media thickness, left ventricular hypertrophy, coronary heart disease, chronic kidney disease and end-stage-renal-disease [7, 16, 17]. Modifiable factors such as smoking, being sedentary, obesity, dyslipidaemia, and dietary issues are associated with development of prehypertension, although it is unclear how these factors contribute in progression from prehypertension to hypertension [6, 13, 18]. Waist-Hip ratio (WHR), body mass index (BMI) and weight-height ratio (WHtR) have all been linked with development of prehypertension and full-blown hypertensive disease for all ages [19–22].

The cost of screening and treating a hypertensive individual in Kenya is USD 178 monthly [23]. This is worrying in an economy where while 25% are hypertensive and therefore likely to

spend heavily on treatment, 36% of them live on <1 USD a day [23, 24]. Currently, first line management of prehypertension is lifestyle change targeting modifiable factors as opposed to pharmacologic interventions, unless there is concurrent diabetes, kidney, or cardiac disease [4, 9]. Evidence supporting pharmacological intervention is inconclusive [25]. Exercise independently lowers BP in both hypertensive and non-hypertensive individuals [10, 26, 27]. Aerobic exercise in younger individuals or those with prehypertension not only lowers BP but also mitigates occurrence of full blown hypertensive disease even independently of dietary adjustment and weight loss interventions. Further, it has better results in all forms of hypertension where drug-therapy has failed [7, 9, 28, 29].

The search for a feasible way to prevent transition of prehypertension to hypertension therefore is necessary. We recently found that moderate-intensity exercise regimes involving bouts of <10 minutes but whose cumulative weekly time equals current World Health Organization (WHO) recommendations of 150 minutes has higher appeal and, yet, confer appreciable health benefits on sedentary normotensive individuals aged ≥50 years [30–32]. Existing guidelines of moderate intensity exercise for adults have traditionally been achieved by performing 30–60 minute bouts of exercise for 3–5 days weekly, and, for hypertension, there is advocacy to do this daily [27, 32, 33]. Despite our recent findings on beneficial health outcomes of shorter bouts, [30, 31], it is unclear if these benefits would translate similarly for sedentary individuals with prehypertension. Longer regimes of ≥30 minutes in 3–5 days weekly as currently recommended are beneficial but lack appeal [27, 34]. Studies on optional regimes are scanty, with pockets of emerging literature showing accumulating short exercise bouts may impact BP. Emerging knowledge points that accumulating running time of 30 minutes daily, in frequent short bouts of ≥10 minutes lowers SBP in non-hypertensive individuals in 24 hours to a few days [35]. This, however, not only remains inconclusive, but evidence/literature on longer term effect of short-bouts on BP is lacking. Further, evidence on effect of short-intermittent exercise on DBP is minimal, but in a study using 10-minutes-walking exercise tests reduced SBP but not DBP [36]. We are not aware of any randomized-controlled trial that has examined the effect of sub-10 minutes' moderate intensity exercise on sedentary individuals with prehypertension, a neglected subpopulation.

The current work tested for equally or more appealing exercise regimes amongst individuals with prehypertension as a way to control their BP. We evaluated adherence to and BP benefits of cumulative fixed bouts of 7.5-minutes' moderate intensity exercises (F-7.5m-MIE) performed throughout the day compared to the single-continuous 30–60 minute bouts among sedentary adults with prehypertension.

## Methods

### Ethics statement

This work was approved by Moi Teaching and Referral Hospital (MTRH) / Moi University Institutional Research Ethics Committee on 17[th] March 2020 (approval no. 0003551) (S1 Text), received national licensure on 15[th] May 2020 (NACOSTI/P/20/4938) (S2 Text), and availed a physician throughout implementation phase to handle any adverse events. Following explanation of the objectives and procedures of the study, subjects were medically screened and gave written consent to participate in the study.

### Design

This was a randomized controlled field trial amongst residents from western region of Kenya where >80% are sedentary [37]. This followed an improved adherence and appreciable body composition and cardio-metabolic finding in an older cohort [38].

## Trial registration

The authors confirm that all ongoing and related trials for this intervention are registered. This trial was retrospectively registered with Pan African Clinical Trial Registry (www.pactr. org) (no. PACTR202107584701552) (S3 Text). The investigators earlier felt that being a behavioral as opposed to substance/drug trial, registration would not have been necessary. Upon learning that it met the WHO definition of a clinical trial being a prospective study that assigned participants to behavioural interventions and measuring associated effects, and with the trial already done, retrospective registration became necessary and letter S4 Text was obtained.

## Study population and sampling

We studied 665 sedentary adults who had prehypertension ($\geq$18 (range 18–79) years; weekly metabolic equivalent minutes (MET-minutes) <600; SBP $\geq$120–139 mmHg and/or DBP $\geq$80–89 mmHg) following a local print advertisement. For motivation, screened volunteers received a full physical examination. The study had 3 arms each comprising males and females. ANOVA sample computation using expected DBP means of 72.9±1.4 mmHg for participants performing our trial regime (described below) and 72.2±1.8 mmHg for those on the traditional regime as found in our previous data [39], and, 82 mmHg for the non-interventional group expected to maintain baseline values at end point, and, further, a considered drop out estimated at 25% yielded 750 participants. For this computation and using our quoted previous data, DBP gave larger sample than using SBP. We did individual-level randomization for each sex into trial (F-7.5m-MIE) arm EG1, current standard WHO recommendation (30–60 min bouts) arm EG2, and the non-intervention group CG (no guidelines exist for prehypertension care). After signing an informed consent, participants picked sealed envelopes they personally shuffled, randomly grouping themselves. Thereafter, each was explained to what their regime entailed. The associated progress from recruitment of participants and through all phases for the follow-up period is shown in the flow chart in Fig 1.

## Protocol description

EG1 participants performed 3 bouts of F-7.5m-MIE each daily, their weekly cumulative exercise time reaching 150 minutes. Correspondingly, EG2 participants performed current recommendation of 30–60 minutes' sessions for 3–5 days weekly, similarly yielding 150 minutes. The prescribed home-based moderate intensity exercises (jogging and related activities as per WHO guidelines) were such that they raised heart rate (HR) to 50–70% of participant's maximal expected HR (220 –(minus) age in years, or where one could talk while performing but not sing) [33]. This was performed for 12 weeks. Participants wore activity monitors (Vivoactive 4 Garmin smart watches, Garmin International Inc 1200 East 51st Street, Olathe, Kansas, 66062 USA) on select days but additionally kept exercise logs analysed for adherence and quality control weekly, and, additionally, for data safety and monitoring board reporting and assessments, and for discontinuation and/or medical referral if indicated. Phone calls and text reminders for exercise performance follow-up were done weekly to ensure observance of prescriptions thus reducing attrition and cross-over effects of adopting non-prescribed exercise regimes. Participants meeting 150 minutes of exercise weekly and whose BP dropped or, at worst, remained in prehypertension ranges were retained in the follow-up. Those whose BP rose to hypertensive levels were reviewed by study physician and referred for pharmacotherapy. The CG continued normal lifestyles but were similarly followed up for BP measurements alongside intervention arms after 12 weeks.

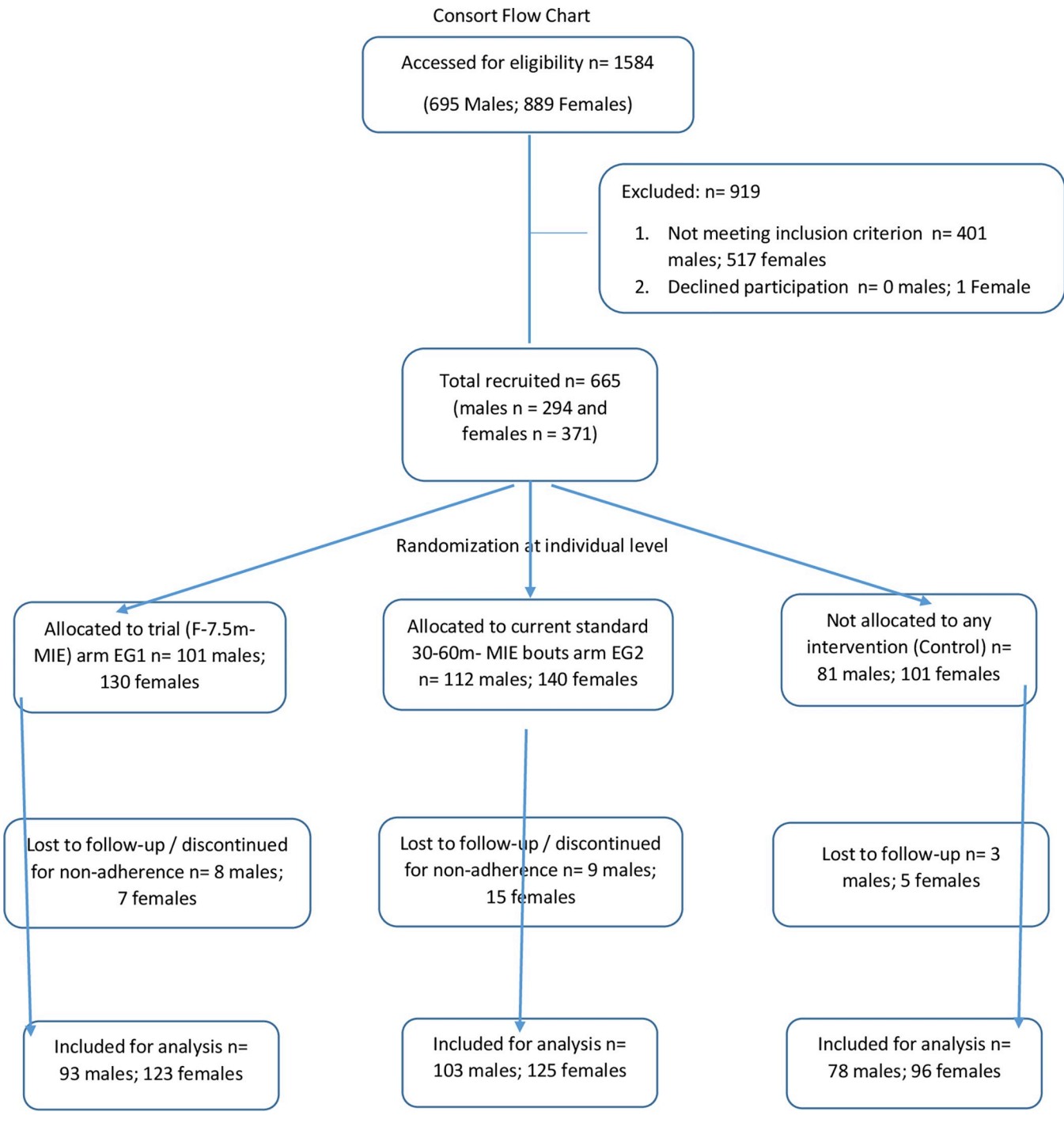

**Fig 1. Participants recruitment and follow up.**

## Data collection

The first participant in the current trial was recruited on 31st August 2020 and the follow-up for the last participant ended on 21st January 2021. Baseline data collected included bio-demographic characteristics, exercise patterns and HR and BP recordings (described below) using Omron M2 Basic (HEM-7120-E) automatic BP monitor (Omron Healthcare Co. Ltd, Kyoto,

Japan). We also measured weight using a mechanical scale (CAMRY, BR9012, Shanghai, China), height, and hip and waist circumferences using a tape measure (also described below). The HR, BP, weight, hip and weight circumferences were repeated after 12 weeks.

## Measures

The Omrom M2 BP monitor described was used for HR and BP measurements. For both, measurements were taken as the average of two recordings done 2 minutes apart with participants awake and in a sitting position, and at rest. The cuff was held at the mid-upper arm and at level with the heart, and a normalization measurement that was discarded initially done to test the cuff and allay anxiety for the participant. The next two recorded measurements were adopted as per guidelines provided by American Heart Association [40]. From their average, we got the HR and BP measurements used for this study. Body composition measurements were taken in centimetres with participants upright standing, with feet positioned close together and arms at the side as provided in the criterion defined by WHO [41]. Specifically, height was taken with participants without shoes and facing straight ahead. This was done from the highest point of the head to the plantar of the foot. Where they had to have shoes, the shoe-sole width was subtracted from final height. For waist circumferences, the tape was run in direct contact to skin at the umbilicus level or point that yielded the least circumference. The widest portion of the buttocks was used for hip circumference. Weight was measured in kilograms with participants on light clothing and without shoes. The WHtR and WHR were computed through dividing the waist circumference by the height and by the hip circumferences respectively. The BMI was computed by dividing the weight in kilograms with the height in metres squared. The pulse pressure (PP) was gotten as the difference between SBP and DBP.

## Analysis

Data were analysed using STATA v.13 at univariate (means and standard deviations) for baseline and week 12, and bivariate level (t-tests; ANOVA) comparing data between groups. Multivariate analysis (mixed type MANOVA; RM ANOVA) comparing data between and within groups (repeated measurements) was also performed. For effect sizes, Cohen's "d" was computed between groups. A $P$ value of $\leq 0.05$ signified a difference in BP between and within groups.

## Results

Male (n = 294) and female (n = 371) participants had mean age of 35.3±12.2 and 34.4±11.8 years respectively. For the males, 39% and 50% had highest level of education as secondary and tertiary respectively. For the females, this was 42% and 35% respectively. For the males and females respectively, 38.7% and 20.6% were in formal employment, 35.3% and 49.6% were in business or farming, and 25.7% and 21.4% were still in college while 8.5% of the females were housewives.

For the males and females respectively, 97.3% (n = 294) and 94.9% (n = 371) had mean SBP $\geq$120 mmHg (129.9±5.4 mmHg and 128.7±4.9 mmHg) at baseline, and 83.7% of the males (n = 294) and 83% of the females (n = 371) had DBP $\geq$80 mmHg (84.0±3.4 mmHg and 84.1 ±3.3 mmHg respectively). Analysis of variance showed SBP at baseline was not different among males and females allocated into the different exercise regimes (p = 0.48 and p = 0.40 respectively). Similarly, DBP was not different for the males and females in the different regimes at baseline (p = 0.90 and p = 0.88 respectively). Among the males, BMI, WHR, WHtR, PP and HR means were 24.2±3.8, 0.92±0.08, 0.49±0.07, 98.3±4.0 and 74.1±9.6 beats per minute (bpm) respectively. For the females, the same, respectively, were 25.6±4.7, 0.91±0.08, 0.52

**Table 1. Baseline bio-demographic characteristics of the participants.**

|  | M_EG1 (n = 101) | M_EG2 (n = 112) | M_CG (n = 81) | F_EG1 (n = 130) | F_EG2 (n = 140) | F_CG (n = 101) |
|---|---|---|---|---|---|---|
| Age (years) | 35.8±12.8 | 34.5±11.9 | 35.7±12.2 | 35.0±12.8 | 34.1±11.3 | 34.2±11.1 |
| WHtR | 0.49±0.06 | 0.49±0.06 | 0.51±0.07 | 0.53±0.08 | 0.53±0.08 | 0.52±0.07 |
| WHR | 0.92±0.08 | 0.93±0.09 | 0.92±0.06 | 0.90±0.08 | 0.90±0.09 | 0.92±0.07 |
| BMI (Kg/M$^2$) | 23.8±3.4 | 24.2±3.8 | 24.8±4.22 | 26.2±4.9 | 25.9±4.8 | 24.5±4.3 |
| SBP (mmHg) | 129.5±6.1 | 130.0±5.7 | 129.0±5.4 | 128.6±5.8 | 127.8±5.7 | 127.6±5.6 |
| DBP (mmHg) | 82.5±5.1 | 82.7±4.7 | 82.7±3.8 | 82.8±5.0 | 82.5±4.9 | 82.5±4.3 |
| Pulse Pressure | 98.1±4.4 | 98.5±4.2 | 98.1±3.2 | 98.0±4.2 | 97.6±3.9 | 97.5±3.4 |
| Pulse rate (b/m) | 75.2±9.2 | 73.1±10.0 | 74.2±9.5 | 77.5±11.4 | 77.0±9.2 | 77.5±11.9 |

Data presented as mean±SD. WHtR: Waist-Height ratio, WHR: Waist Hip Ratio, SBP: systolic blood pressure, DBP: diastolic blood pressure, mmHg: millimetres of mercury, b/m: beats per minute, Groups: M_EG1: short bouts, male; M_EG2: long bouts, male; M_CG: no intervention males; F_EG1: short bouts, female; F_EG2: long bouts, female; F_CG: no intervention females.

±0.08, 97.8±3.89 and 77.3±10.7. Baseline demographic characteristics based on randomised groups are presented in Table 1.

For the follow-up period, activity data were similar for the two interventional groups with cumulative exercise minutes 155.6+2.9 versus 154.5+3.2 minutes (M_EG1 and M_EG2 respectively) for the males (p = 0.1). Similarly, among the females, it was 154.6+3.1 versus 154.7+3.1 minutes for F_EG1 and F_EG2 respectively (p = 0.70). For the overall adherence, 92.9% of the participants completed the 12 weeks follow-up with 92.1% in M_EG1, 92% in M_EG2 and 96.3% in M_CG for the male groups and 94.6% in F_EG1, 89.3% in F_EG2 and 95% in F_CG for the female groups.

After 12 weeks adherence to prescribed exercises for all groups (M_EG1 n = 93, M_EG2 n = 103, M_CG n = 78, F_EG1 n = 123, F_EG2 n = 125, F_CG n = 96), there were varied effects on the cardiovascular and body composition measurements as shown in Table 2.

Regression models for differences in the change between baseline and endpoint for the various groups all for males and females respectively showed that SBP (p<0.001; F = 31.39 and p<0.001; F = 41.90), DBP (p<0.001; F = 18.44 and p<0.001; F = 34.45), WHtR (p = 0.18; F = 1.78 and p = 0.69; F = 0.16), WHR (p = 0.01; F = 6.83 and p = 0.44; F = 0.60), BMI (p = 0.28; F = 1.16 and p = 0.45; F = 0.58), HR (p<0.001; F = 9.42 and p<0.001; F = 10.72) and PP (p = 0.01; F = 6.47 and p = 0.11; F = 2.56) all yielded mixed outcomes.

At baseline, more participants had higher values for various cardiovascular and body composition than observed at endpoint. Fig 2 shows the percentage dropping to respective variables' cut-offs after the 12 weeks follow-up. It is noteworthy that the mean values changed as shown in Table 2 even where percentage drop of participants to below cut-offs for the various variables was marginal. The change difference for the two interventional groups were also minimal, with Cohen's d for effect sizes for the various variables between baseline and week 12 for males and females respectively being: WHtR (0.02 and 0.2), WHR (0.1 and 0.3), BMI (0.04 and 0.2), SBP (0.2 and 0.01), DBP (0.03 and 0.1), HR (0.1 and 0.2) and pulse (0.05 and 0.1).

## Discussion

All the 665 participants of the current study had prehypertension at baseline, and were sedentary as per the WHO global physical activity questionnaire. The fact that majority had at least secondary level education made it easier to communicate what having prehypertension meant, the importance of exercise interventions, the prescribed exercise instructions, and how to monitor exercise activity. Similarly, the participants were predominantly young adults which

**Table 2. Cardiovascular and body composition measurements between weeks 0 and 12.**

| Variable | Group | Week 0 | Week 12 | Mean Δ (wk12-wk0) | P value (Δ from wk 0) |
|---|---|---|---|---|---|
| Male | | | | | |
| WHtR | M_EG1 | 0.49±0.06 | 0.48±0.06 | -0.01±0.02 | 0.00 |
| | M_EG2 | 0.48±0.06 | 0.48±0.06 | -0.01±0.02 | 0.00 |
| | M_CG | 0.51±0.07 | 0.51±0.07 | 0.00±0.02 | 0.82 |
| WHR | M_EG1 | 0.92±0.08 | 0.90±0.08 | -0.01±0.03 | 0.00 |
| | M_EG2 | 0.93±0.09 | 0.91±0.12 | -0.02±0.1 | 0.00 |
| | M_CG | 0.92±0.06 | 0.93±0.06 | 0.01±0.05 | 0.06 |
| BMI (Kg/M$^2$) | M_EG1 | 24.1±3.4 | 23.7±3.4 | -0.4±0.7 | 0.41 |
| | M_EG2 | 24.0±3.8 | 23.5±3.6 | -0.5±0.7 | 0.99 |
| | M_CG | 24.6±3.7 | 24.7±3.7 | 0.1±0.5 | 0.00 |
| SBP (mmHg) | M_EG1 | 129.7±6.1 | 123.1±9.2 | -6.6±8.0 | 0.00 |
| | M_EG2 | 129.9±5.6 | 121.9±9.5 | -8.0±8.3 | 0.00 |
| | M_CG | 129.3±5.1 | 129.9±5.7 | 0.6±6.1 | 0.08 |
| DBP (mmHg) | M_EG1 | 82.3±5.1 | 77.0±5.5 | -5.3±6.8 | 0.04 |
| | M_EG2 | 82.6±4.5 | 77.5±5.9 | -5.1±6.5 | 0.01 |
| | M_CG | 82.6±3.8 | 81.7±5.4 | -0.9±5.7 | 0.31 |
| Pulse | M_EG1 | 98.1±4.4 | 92.4±6.1 | -5.7±6.7 | 0.70 |
| Pressure | M_EG2 | 98.4±4.1 | 92.3±6.4 | -6.1±6.1 | 0.79 |
| | M_CG | 98.2±3.2 | 97.8±4.5 | -0.4±4.8 | 0.97 |
| Pulse (b/m) | M_EG1 | 75.1±9.1 | 72.2±7.3 | -2.9±6.4 | 0.00 |
| | M_EG2 | 72.7±9.9 | 70.1±7.9 | -2.6±6.1 | 0.00 |
| | M_CG | 73.8±9.4 | 75.1±16.6 | 1.2±12.6 | 0.00 |
| Female | | | | | |
| WHtR | F_EG1 | 0.53±0.08 | 0.52±0.07 | -0.01±0.02 | 0.00 |
| | F_EG2 | 0.52±0.08 | 0.52±0.07 | -0.01±0.02 | 0.00 |
| | F_CG | 0.51±0.07 | 0.51±0.07 | 0.00±0.03 | 0.00 |
| WHR | F_EG1 | 0.91±0.08 | 0.9±0.08 | -0.02±0.04 | 0.00 |
| | F_EG2 | 0.9±0.09 | 0.88±0.08 | -0.02±0.03 | 0.00 |
| | F_CG | 0.92±0.07 | 0.91±0.11 | -0.01±0.1 | 0.04 |
| BMI (Kg/M$^2$) | F_EG1 | 26.3±4.9 | 25.7±4.6 | -0.6±0.9 | 0.01 |
| | F_EG2 | 25.5±4.5 | 25.1±4.0 | -0.5±1.2 | 0.06 |
| | F_CG | 25.5±4.1 | 24.8±4.1 | 0.3±0.7 | 0.92 |
| SBP (mmHg) | F_EG1 | 128.6±5.7 | 121.3±8.4 | -7.3±6.5 | 0.00 |
| | F_EG2 | 127.7±5.7 | 120.5±7.5 | -7.2±7.5 | 0.01 |
| | F_CG | 127.7±5.4 | 127.0±7.7 | -0.7±6.9 | 0.00 |
| DBP (mmHg) | F_EG1 | 82.9±4.8 | 77.4±5.5 | -5.5±5.9 | 0.12 |
| | F_EG2 | 82.5±4.8 | 77.4±6.0 | -5.1±6.1 | 0.01 |
| | F_CG | 82.4±4.3 | 82.1±5.9 | -0.3±6.5 | 0.26 |
| Pulse | F_EG1 | 98.1±4.2 | 92.0±5.6 | -6.1±5.3 | 0.16 |
| Pressure | F_EG2 | 97.6±3.9 | 91.8±5.6 | -5.8±5.6 | 0.48 |
| | F_CG | 97.5±3.5 | 97.1±5.7 | -0.4±5.9 | 0.35 |
| Pulse (b/m) | F_EG1 | 77.4±11.0 | 73.3±8.5 | -4.0±6.4 | 0.00 |
| | F_EG2 | 77.1±9.4 | 74.1±7.7 | -3.0±5.2 | 0.00 |
| | F_CG | 77.0±11.0 | 75.8±9.3 | -1.2±7.3 | 0.00 |

WHtR: Waist-Height Ratio; WHR: Waist-Hip Ratio; BMI: Body Mass Index; Kg/M$^2$: Kilogram per metre squared; SBP: Systolic Blood Pressure; DBP: Diastolic Blood Pressure; mmHg: millimetres of mercury; b/m: beats per minute. M_EG: Male Experimental Group; F_EG: Female Experimental Group; CG: Control Group. All values are Means±SD. P<0.05: significant difference in mean change between the two regimes.

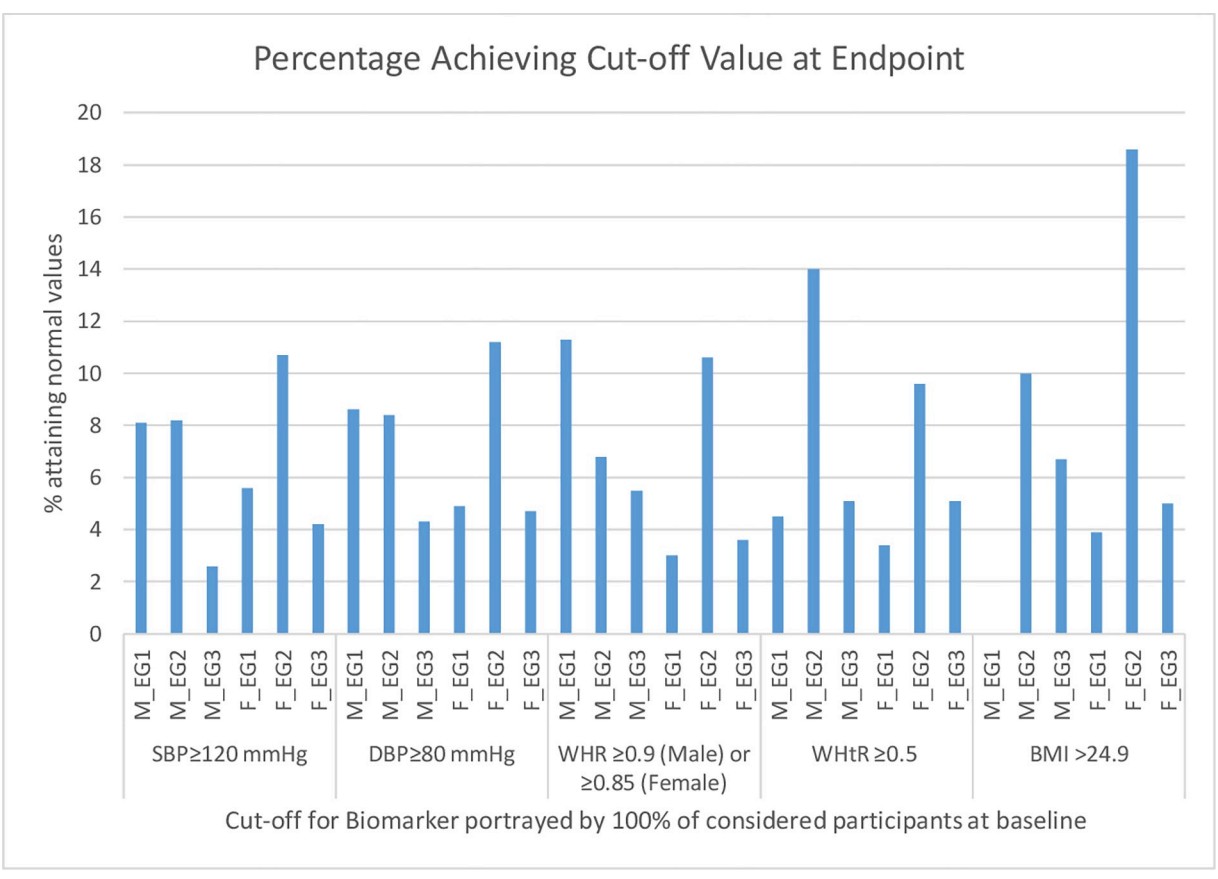

**Fig 2. Percentage attaining recommended cut-offs after 12 weeks.**

allowed ease in exercise participation. However, almost all participants were in formal employment, in business or still in college and therefore had to purposely make time to follow their respective exercise prescriptions.

After the 12 weeks follow-up, those retained in the intervention groups had similar cumulative exercise time when groups per sex were compared. This allowed comparisons of the various variables yielded. Among the males from the two interventional groups, adherence rates were similar while females on shorter bouts of exercise had a 5-percentage-points higher adherence compared to those in the longer bouts' group. Previous work on similar exercise regimes from the same setting have equally shown higher adherence in both males and females performing shorter bouts in an older population, suggesting that such regimes may be more appealing [30]. The current study equally suggests that across all adult ages, adoption of shorter exercise bouts may improve exercise adherence since individuals maintaining their exercise prescriptions matched those performing the WHO standard exercise regimes. Where individuals have prehypertension like in the current study, this may yield more benefits in control of their BP.

At the end of 12 weeks for body composition measurements, there was a significant reduction in both WHtR and WHR in the two experimental groups for both sexes compared to the baseline. The manner of this reduction was similar for the two regimes when compared between themselves and with the control group. To provide for body fuel during the prescribed exercises, the body mainly breaks down fatty stores. When these fats are in the

abdominal region, and coupled with fact that height minimally changes over such a short follow-up period, and, further, that the hip region is mainly muscular as opposed to fatty tissue, the breakdown of the waist-region fats from the two interventions lowered these ratios. For BMI among males, recorded reduction was insignificant for both interventional groups but noteworthy was that the control group had a significant increase. Females in the short-bouts arm had a reduction in their BMI but the changes in the longer bouts and the control groups were insignificant. It is likely that the 12 weeks follow-up period could not allow clear and appreciable changes as weight reduction which affects BMI has been shown to be slower than other anthropometric measurements such as waist circumference change that affects WHR and WHtR, explaining the differences observed in these measures [42]. We previously observed positive changes in body composition from the same setting on healthy-sedentary individuals by use of comparable exercise regimes [31]. Elsewhere, similar follow-up periods for accumulated shorter bouts of exercise have yielded favourable body compositions [43, 44], although they were not specific for individuals with prehypertension.

There were mixed outcomes regarding cardiovascular effects of the 12 weeks' exercise interventions. For both males and females, there was significant reduction in SBP for both experimental groups but not for the control. The manner of this reduction was similar for both regimes. The DBP also dropped in both regimes for the males and in the longer bouts for the female, and there was no change in the control group. There was a reduction of the resting HR by 3 to 4 BPM for the females and 2.5 to 3 BPM for males in all regimes, with no difference inter-regimes. For males and females, the 12 weeks follow-up reduced PP by 6 mmHg for both exercise regimes, although the difference was not significant. While these show that the prescribed exercises improved these cardiovascular measures, probably a longer follow-up is necessary to reduce this further. Still, the absolute PP reduction alludes to the narrowing of the pressure difference between SBP and DBP, which is an independent risk factor for cardiovascular disease related to stiffening of blood vessels [45]. In this study, short bouts of aerobic exercises reduced this stiffening in a similar manner to the currently prescribed bouts. A recent meta-analysis showed that there was no difference between accumulated shorter bouts and the traditionally longer-continuous bouts of exercise in BP modulation in the general population [46]. The present work shows similar benefits among a sub-population with prehypertension.

When the mean change in value of the various variables between baseline and endpoint was modelled for difference between regimes, there were significant differences observed between the two interventional groups on the one hand and the control group on the other. For SBP, DBP and HR, all had differences in mean change between the two interventional groups for both sexes. For WHR and PP, there a difference between the short and the long bouts among the males only. In both males and females, there was no observable difference between the two regimes for BMI and WHtR mean change. When we considered the effect sizes in the differences between the two regimes, computed Cohen's "d" seemed to suggest that variances were negligible since the effects sizes were mostly below 0.2. Considering the outcomes, the two interventional regimes were largely similar showing that the trial regime was comparable to the existing WHO standard for moderate intensity exercises as currently recommended. The mechanisms in body composition and cardiovascular measurements change following an F-7.5m-MIE is somewhat same as that in the longer regime. Previous studies from our set up and elsewhere have shown that shorter prescribed bouts could actually yield better outcomes on body composition and cardiovascular measures [30, 31, 39, 47], which differences, although apparent but minimal in the current work, could not be pursued further for direction because our design could not support testing for superiority.

The percentage drop for WHR means $\geq 0.9$ and $\geq 0.85$ for males and females respectively was higher for short and long bouts for males and females respectively, but these differences

were marginal. A similar mixed picture was observed for WHtR >0.5 and also for BMI but here, the longer regime appeared superior in the percentage drop in both categories for both males and females. These observations are replicated from an older population in the same set up [31], but one who although sedentary were not prehypertensive. We were unable to find any studies that have looked at effect of such short exercise regimes on similar cardiovascular and body composition variables as ours, and the current attempt is the first we are aware of. Still, probably a longer follow-up would illuminate these differences better, and, additionally, reduce those remaining with prehypertensive BPs or with body composition measurements above their respective cut-offs by an even larger proportion for both sexes and regimes. Of importance is that even with these mixed results, all variables had mean values drop between baseline and endpoint, suggesting similar value in the two experimental regimes. Evidence from previous studies on the effect of short and long bouts of exercises on body composition is inconclusive. While Alizadeh et al. [48] showed significant reduction in BMI and weight among females on shorter bouts of exercises compared to the long bouts of exercises, other studies [49–51] found both intermittent short bouts and continuous exercise programs to be effective in weight loss and improving body composition with no significant difference between the programs. On the contrary, a study involving middle aged obese women [52] showed that long bouts exercise are superior in the reduction of BMI, weight and fat mass. Further, Chung et al. [52] concluded that multiple short bout exercises are better than prolonged exercise when the goal is to reduce waist circumference. However, a meta-analysis on the effects of continuous compared to accumulated exercise on health showed no statistical differences between short and long exercise for any anthropometric or body composition outcome except body weight [46]. Further studies on the long term effect of short bouts exercises on body composition are recommended.

In the current study, both the short and the long exercise regimes had at least an 8 percentage drop in males who, separately, had SBP $\geq$120 and DBP $\geq$80 mmHg at baseline. Among the females, the percentage drop was higher among the long bouts' group than in the shorter for SBP, but similar at 8.5% for DBP. This underscores the similarity in effect for the two exercise regimes among individuals needing to regulate their BPs. While the two regimes had similar percentage drops for SBP and DBP in males and also DBP in females, it was unclear from the current study why this differed in SBP for the females. While we are not aware of any study that has looked at effect of aerobic exercises among individuals with prehypertension that lasted a similar period, one study has found regular moderate-intensity exercises lasting 10 minutes per session have yielded BP reductions of up to 5 mmHg [35]. High intensity exercises performed for a period slightly longer than in the current study have shown SBP and DBP decreases by about 8.7 mmHg and 5.4 mmHg respectively, similar to values in our current moderate-intensity study [53]. This is further supported by a systematic review and meta-analysis of randomized trials that showed comparable BP changes between high and moderate intensity individuals with pre- to established hypertension [54]. Given that in BP control adherence to prescriptions of moderate intensity exercises is higher than that for high intensity exercises [55], and, further, now that shorter regimes of moderate intensity exercises are comparable to the longer regimes of similar intensity in adherence, then prescriptions of shorter-bouts moderate intensity exercises here demonstrated as similarly beneficial in BP modulation for individuals with prehypertension become an important intervention for this sub-population.

Given that the drop-outs in each arm were less than 30% as the set criterion for follow-ups exceeding 4 weeks [56], and, further, the drop in SBP exceeding 7 mmHg in 12 weeks for the two interventional groups as has been shown elsewhere using higher intensities and for longer periods [53], and this a significant drop when compared with the control group, we consider

the current study a success. It demonstrates that F-7.5m-MIE performed thrice daily and whose cumulative exercise time reaches the 150 minutes mark as currently advocated for by WHO to be achieved in bouts of 30–60 minutes is equally beneficial among individuals with prehypertension. Exercise prescriptions involving shorter regimes which could be more appealing for some could play a crucial role in prevention of full-blown hypertension for individuals who have prehypertension. The shorter bouts in the current study had an insignificantly higher adherence rate but probably in a longer follow-up as demonstrated in our previous studies using normotensive individuals [31] or in a cross-over design, this would differ more significantly and/or shed more light. With the shorter bouts easier to implement, adhere to and yiedling similar cardiovascular disease protection as the longer regimes in sedentery individuals who have prehypertension, they should be considered in quest to reduce progression of prehypertension to hypertension, and, thus, providing more feasibe options that positively impact outcomes associated with exercise prescriptions practice. The current study therefore offers an additional yet equally appealing exercise regime that individuals with prehypertension could chose from and adopt in mitigating progression to full-blown hypetensive states.

## Limitations

Blinding of participants was impossible in the study and as such, interventional group participants may have had peer interactions that affected adherence, and likely also influenced the control group. Dietary records, smoking and use of non-medical drugs/stimulants, known confounders of BP and body compositions were not controlled for. Additionally, activity monitors were limited for the 665 participants and therefore not available for everyone throughout the follow-up period, with the control group completely missing out. These limitations may have affected the quality of our data and this may affect the generalisability of the current results.

## Conclusion

In sedentary individuals with prehypertension, F-7.5m-MIE performed over 12 weeks reduce BP and improve body composition to a similar magnitude as do the traditional longer bouts that last 30–60 minutes as currently recommended by WHO. The high adherence to this shorter bouts' regime offer additional prescriptive options that could be used for individuals who are sedentary and have prehypertension.

## Supporting information

**S1 Data. Data.**
(XLSX)

**S1 Trial protocol. Trial protocol.**
(PDF)

**S1 Text. IREC approval.**
(PDF)

**S2 Text. Research permit.**
(PDF)

**S3 Text. Clinical trials registration number.**
(PDF)

**S4 Text. Trial approval letter.**
(PDF)

**S5 Text. Reporting guidelines.**
(PDF)

## Author Contributions

**Conceptualization:** Karani Magutah, Grace Mbuthia, James Amisi Akiruga.

**Data curation:** Karani Magutah.

**Formal analysis:** Karani Magutah, Grace Mbuthia.

**Funding acquisition:** Karani Magutah.

**Investigation:** Karani Magutah.

**Methodology:** Karani Magutah, James Amisi Akiruga, Diresibachew Haile.

**Project administration:** Karani Magutah.

**Resources:** Karani Magutah.

**Software:** Karani Magutah.

**Supervision:** Karani Magutah.

**Validation:** Karani Magutah.

**Visualization:** Karani Magutah.

**Writing – original draft:** Karani Magutah.

**Writing – review & editing:** Karani Magutah, Grace Mbuthia, James Amisi Akiruga, Diresibachew Haile, Kihumbu Thairu.

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
