## [Decision Letter · Decision Letter 0]

7 Mar 2022

PGPH-D-21-00205

The Effect of Fixed 7.5 minutes’ Moderate Intensity Exercise bouts on Body Composition and Blood Pressure among Sedentary Adults with Prehypertension in Western-Kenya.

Dear Dr. Magutah,

Thank you for submitting your manuscript to PLOS Global Public Health. After careful consideration, we feel that it has merit but does not fully meet PLOS Global Public Health’s publication criteria as it currently stands. Therefore, we invite you to submit a revised version of the manuscript that addresses the points raised during the review process.

EDITOR'S Comments: Following the comments of the reviewers, the Editor is request special attention to the following areas of concerns;

The abstract has been restructured and we would request you read through and compare with the original to ensure that it reflects concepts and results in the original manuscript,Kindly describe the measures applied in the study in some appreciable details as raised by reviewer 1. It would be appropriate in the section of the methods, under a sub-heading "measures", to describe who the constructs where measured and scales used and computational derivations, example BMI among others. Similarly, in a separate sub-heading "data collection procedures" .Interestingly, computing the Effect Sizes of the outcomes would provide more information about the impact of the study as raised by the reviewer. This can be be compared between Experimental Group 1 and Experimental Group 2.Points to remember when revising the discussion as recommended: Brief review of the main objective for the study and rationale, provide information whether the study answered the major  research questions raised by the problem phenomenon, if so in what ways and explain dynamics? What is observed as reasons for the type of outcomes in the study? Are there implications for practice? What contributions has the study made?A minor issue to be considered is removing "The" from the title of the study and organizing the analyzed data to show what has been used to derive final measures.

We look forward to receiving your revised manuscript.

Kind regards,

Nnodimele Onuigbo Atulomah, PhD

Academic Editor

Journal Requirements:

1. Please amend your detailed Financial Disclosure statement. This is published with the article, therefore should be completed in full sentences and contain the exact wording you wish to be published.

i). State the initials, alongside each funding source, of each author to receive each grant.

ii). State what role the funders took in the study. If the funders had no role in your study, please state: “The funders had no role in study design, data collection and analysis, decision to publish, or preparation of the manuscript.”

2. Please ensure that the funders and grant numbers match between the Financial Disclosure field and the Funding Information tab in your submission form. Note that the funders must be provided in the same order in both places as well.

3. Please update your Competing Interests statement. If you have no competing interests to declare, please state: “The authors have declared that no competing interests exist.”

4. In the online submission form, you indicated that your data will be submitted to a repository upon acceptance. We strongly recommend all authors deposit their data before acceptance, as the process can be lengthy and hold up publication timelines. Please note that, though access restrictions are acceptable now, your entire data will need to be made freely accessible if your manuscript is accepted for publication. This policy applies to all data except where public deposition would breach compliance with the protocol approved by your research ethics board. If you are unable to adhere to our open data policy, please kindly revise your statement to explain your reasoning and we will seek the editor's input on an exemption. Please be assured that, once you have provided your new statement, the assessment of your exemption will not hold up the peer review process.

5. We ask that a manuscript source file is provided at Revision. Please upload your manuscript file as a .doc, .docx, .rtf or .tex. If you are providing a .tex file, please upload it under the item type ‘LaTeX Source File’ and leave your .pdf version as the item type ‘Manuscript’.

6. We ask that a manuscript source file is provided at Revision. Please upload your manuscript file as a .doc, .docx, .rtf or .tex. If you are providing a .tex file, please upload it under the item type ‘LaTeX Source File’ and leave your .pdf version as the item type ‘Manuscript’.

7. Please include your tables as part of your main manuscript and remove the individual files. Please note that supplementary tables should remain as separate “Supporting Information” files.

8. We noticed that you have two Figure 1's in your manuscript. Please update your figure numbers and cite them accordingly.

9. Please provide separate figure files in .tif or .eps format only and ensure that all files are under our size limit of 20MB.

10. We have noticed that you have uploaded supporting information but you have not included a list of legends. Please add a full list of legends for all supporting information files (including figures, table and data files) after the references list.

Additional Editor Comments (if provided):

The reviewers have made pertinent observations in the manuscript requiring your attention to strengthen the final draft to be made. Similarly, I have made significant editorial changes to the abstract which will accompany these comments in an attachment you should download and consider carefully. If it reflects what your paper is expressing you may adopt. I would strongly recommend adopting the draft.

Reviewers' comments:

Reviewer's Responses to Questions

**Comments to the Author**

1. Does this manuscript meet PLOS Global Public Health’s publication criteria? Is the manuscript technically sound, and do the data support the conclusions? The manuscript must describe methodologically and ethically rigorous research with conclusions that are appropriately drawn based on the data presented.

Reviewer #1: Yes

Reviewer #2: Yes

2. Has the statistical analysis been performed appropriately and rigorously?

Reviewer #1: Yes

Reviewer #2: Yes

3. Have the authors made all data underlying the findings in their manuscript fully available (please refer to the Data Availability Statement at the start of the manuscript PDF file)?

Reviewer #1: No

Reviewer #2: No

4. Is the manuscript presented in an intelligible fashion and written in standard English?

Reviewer #1: No

Reviewer #2: Yes

5. Review Comments to the Author

Reviewer #1: This is an informative analysis of the effects of short-bouts and long-bouts exercises on body composition and BP. I think this analysis is generally done well. However, I think the presentation of the methodology, results, and discussion could be improved for clarity. Therefore, my suggestions are mainly minor and detailed below:

Key comments

It would be important for the authors to describe the measures in detail. I suggest that the authors should describe data collection and measures separately. Currently, there is a mix of 'data collection' and 'measures' descriptions, making it difficult to read. The authors need to clearly state what the 'body composition measures' are (i.e., WHR etc.), and how these were measured/calculated. It may look straightforward but its always good to describe how BMI, WHR, WHtR were calculated. Since the title contains the words' body composition and BP' as the main outcomes, the measures section should clearly explain these.

The data analysis section could be clearer. The authors performed a RMANOVA to report the differences between and within groups. Readers may benefit if authors include the type of RMANOVA used (Mixed effect?). Again, it would be interesting to see the effect sizes of the interventions on the outcomes of interest instead of just mentioning that there were significant differences. Also consider reporting the F-values. Just presenting the p-values does not give enough information.

It is insufficient to just state that the study tested a 'more appealing' exercise regime. Would recommend also adding rationale/backing for why you claim that short bouts of exercises would be more appealing to the older adults.

The discussion should be revised. As it is, the discussion section reads like a repetition of the result section. I do not see the need to present the results in the discussion section i.e., the p-values etc. The results observed in the analysis should be supported with scientific explanation. The authors should at least seek to explain/address 'Why' there was a significant reduction in WHtR and WHR in the two experimental groups (what are the possible mechanisms?). Similarly, it would be important to have a detailed discussion on why short bouts had significant effects on BMI among females. In contrast, long bouts did not yield substantial effects (This is also where showing the effect sizes might be handy as compared to just revealing the p-values).

Minor comments

The first sentence on the abstract needs some revision. i.e., it says, 'Prehypertension is a modifiable risk factor for CVD defined as SBP…' – The definition is for prehypertension (i.e., SBP=>120-139 and DBP=>80-89). However, the way the sentence is constructed could be mistaken as if the authors are defining CVD.

Use the full word at the first mention of any abbreviated word e.g., WHO.

I have problems with using T1, T2 etc. to denote Trial arm 1 and 2, respectively. As this is a follow up study, I kept confusing whether this is time 1 (baseline) or time 2 (endline). I believe other readers may have similar confusion, please consider revising.

In the methodology section, it would be good to clarify the age range of the participants instead of just mentioning that they were older adults. I know authors have referred to previous related work with the age range stated but would be essential to include in this paper's methodology.

Table 2, include footnotes on the meanings of WHtR, WHR, MT1 etc.…

Need to reorganize the figures, the Consort flow chart could be figure 1 and so on to allow for logical flow in what is being described in the methodology/results.

I suggest the authors to present a table summary for the characteristics of the study participants.

There is a need for English language editing of the entire manuscript.

Reviewer #2: The language was lucid and clear. The manuscript presented in an intelligible fashion and written in standard English?

The articles is clear, correct, and unambiguous. Few typographical error were pointed out. typographical or grammatical errors were pointed out if the author do the revision and the specific errors pointed to are corrected.

6. PLOS authors have the option to publish the peer review history of their article (what does this mean?). If published, this will include your full peer review and any attached files.

**Do you want your identity to be public for this peer review?** For information about this choice, including consent withdrawal, please see our Privacy Policy.

Reviewer #1: No

Reviewer #2: No

---

## [Decision Letter · Decision Letter 1]

16 Jun 2022

PGPH-D-21-00205R1

Effect of Fixed 7.5 minutes’ Moderate Intensity Exercise bouts on Body Composition and Blood Pressure among Sedentary Adults with Prehypertension in Western-Kenya.

Dear Dr. Magutah,

Thank you for submitting your manuscript to PLOS Global Public Health. After careful consideration, we feel that it has merit but does not fully meet PLOS Global Public Health’s publication criteria as it currently stands. Therefore, we invite you to submit a revised version of the manuscript that addresses the points raised during the review process.

EDITOR:

Dear Author,

Please perform language checks before this manuscript can be accepted for publication.

The decision of this manuscript is justified based on PLOS Global Public Health’s publication criteria and not on its novelty or perceived impact.

We look forward to receiving your revised manuscript.

Kind regards,

Zulkarnain Jaafar

Academic Editor

Journal Requirements:

Additional Editor Comments (if provided):

Reviewers' comments:

Reviewer's Responses to Questions

**Comments to the Author**

1. If the authors have adequately addressed your comments raised in a previous round of review and you feel that this manuscript is now acceptable for publication, you may indicate that here to bypass the “Comments to the Author” section, enter your conflict of interest statement in the “Confidential to Editor” section, and submit your "Accept" recommendation.

Reviewer #1: All comments have been addressed

Reviewer #2: All comments have been addressed

2. Does this manuscript meet PLOS Global Public Health’s publication criteria? Is the manuscript technically sound, and do the data support the conclusions? The manuscript must describe methodologically and ethically rigorous research with conclusions that are appropriately drawn based on the data presented.

Reviewer #1: No

Reviewer #2: Yes

3. Has the statistical analysis been performed appropriately and rigorously?

Reviewer #1: Yes

Reviewer #2: Yes

4. Have the authors made all data underlying the findings in their manuscript fully available (please refer to the Data Availability Statement at the start of the manuscript PDF file)?

Reviewer #1: No

Reviewer #2: Yes

5. Is the manuscript presented in an intelligible fashion and written in standard English?

Reviewer #1: (No Response)

Reviewer #2: Yes

6. Review Comments to the Author

Reviewer #1: I have had problems to review the paper because authors did not provide a point-by-point response so I had to read through and through to ensure the earlier comments have been addressed and locate that in the manuscript. It seems the authors have done a good job to address my initial comments and I commend them for that. One minor comment is that the manuscript should be thoroughly checked for English language before publication otherwise most of the technical comments were addressed. Another comment in the discussion section Paragraph 2, “…adherence in both males and females performing shorter bouts in an older population, proposing that such regimes appeal more…” I suggest the authors to write “adherence in both males and females performing shorter bouts in an older population, suggesting that such regimes may be more appealing”

- State data availability at the end of the manuscript

Reviewer #2: (No Response)

7. PLOS authors have the option to publish the peer review history of their article (what does this mean?). If published, this will include your full peer review and any attached files.

**Do you want your identity to be public for this peer review?** For information about this choice, including consent withdrawal, please see our Privacy Policy.

Reviewer #1: No

Reviewer #2: **Yes: **Oyerinde Oyewole Olusesan (Ph.D)

---

## [Editor Report · Decision Letter 2]

24 Jun 2022

Effect of Fixed 7.5 minutes’ Moderate Intensity Exercise bouts on Body Composition and Blood Pressure among Sedentary Adults with Prehypertension in Western-Kenya.

PGPH-D-21-00205R2

Dear Dr. Magutah,

We are pleased to inform you that your manuscript 'Effect of Fixed 7.5 minutes’ Moderate Intensity Exercise bouts on Body Composition and Blood Pressure among Sedentary Adults with Prehypertension in Western-Kenya.' has been provisionally accepted for publication in PLOS Global Public Health.

Best regards,

Zulkarnain Jaafar

Academic Editor